# Pharmacokinetics of Novel Crystalline Buntanetap in Mice, Dogs, and Humans

**DOI:** 10.3390/biom15091299

**Published:** 2025-09-10

**Authors:** Alexander Morin, Michael Christie, Eve Damiano, Maria L. Maccecchini

**Affiliations:** Annovis Bio, Inc., 101 Lindenwood Drive, Malvern, PA 19355, USA; christie@annovisbio.com (M.C.); damiano@annovisbio.com (E.D.); maccecchini@annovisbio.com (M.L.M.)

**Keywords:** buntanetap, polymorph, pharmacokinetics, neurodegenerative diseases, blood–brain barrier

## Abstract

Buntanetap is an orally bioavailable small molecule that has been shown to improve cognitive function in patients with Alzheimer’s and Parkinson’s diseases and holds promise for use in other neurodegenerative conditions. Until now, a crystalline anhydrate (Form A) has been used in preclinical and clinical studies. However, a novel dihydrate crystal (Form B) was recently discovered, offering improved solid-state stability without compromising its absorption, systemic exposure, and metabolism. We sought to evaluate the pharmacokinetic (PK) profile of Form B and compare it to the well-characterized PK profile of Form A in a series of studies conducted in mice, dogs, and humans. Our data revealed that although the two forms are distinct and do not interconvert, they exhibit comparable PK profiles both within and across species. Consistent with previous reports, Form A and Form B alike reached fast peak plasma concentrations (<2 h), demonstrated efficient partitioning into brain tissue, and were fully cleared by 12 h post-dose. Furthermore, metabolic profiling showed that both forms produced identical PK profiles for the primary metabolites, N1- and N8-norbuntanetap, confirming that Form B retains the established metabolic characteristics of Form A. These findings support the continued development of Form B for future clinical use, as it combines enhanced solid-state stability with a preserved PK profile essential for buntanetap’s therapeutic efficacy.

## 1. Introduction

Buntanetap (formerly known as posiphen) is an orally bioavailable small molecule which is currently being investigated for the treatment of neurodegenerative disorders such as Alzheimer’s disease (AD) and Parkinson’s disease (PD) [1,2,3]. Its small molecular weight (487.5 Da) and lipophilic properties (partition coefficient, Log D 2.2) allow efficient penetration across the blood–brain barrier (BBB) achieving high brain levels where it exerts therapeutic activity.

Buntanetap is the (+)-enantiomer of phenserine; however, while phenserine has acetylcholinesterase (AChE) inhibiting activity, buntanetap does not [4,5]. Buntanetap has a unique mechanism of action in that it lowers the levels of neurotoxic aggregating proteins by strengthening the binding of iron regulatory protein 1 (IRP1) to the iron-responsive element (IRE), which is located in the conserved RNA stem-loop structure of these proteins [6,7,8,9,10,11]. As a result, the stabilized IRP1-IRE complex prevents the mRNA from binding to the ribosome suppressing the translation of proteins involved in neurodegeneration [12,13]. Specifically, buntanetap has been demonstrated to inhibit the translation of amyloid precursor protein (APP) [2,3,7,10,14,15], alpha-synuclein (αSYN) [10,16,17,18,19,20], tau [3], huntingtin [21], TDP-43 [21], and prion [15].

Buntanetap was originally discovered to only have APP inhibiting activity and no AChE inhibiting (AChEi) activity, in vitro. However, in vivo, whether in rodents, dogs or humans, buntanetap undergoes N-demethylation at N1 and N8 positions, generating N1-norbuntanetap and N8-norbuntanetap, respectively [19]. Then, each metabolite undergoes further N-demethylation producing N1,8-bisnorbuntanetap, which is a minor metabolite in humans and hence omitted from pharmacological evaluation in this manuscript. The first two metabolites have different activities: N1-norbuntanetap has been shown to inhibit the translation of neurotoxic proteins as well as AChE, whereas N8-norbuntanetap exhibits inhibitory activity towards neurotoxic proteins but not towards AChE [3,5,18,19,20,21]. Studies in neuron cultures showed that both N1 and N8 inhibit APP, αSYN, and other neurotoxic proteins [19,20]; hence, it is likely that clinical efficacy is driven by a compound action of buntanetap and its metabolites. At the same time, N1 may be responsible for a dose-limiting effect of the drug at high doses due to its AChEi activity, and thereby observing its low concentrations in humans is crucial, while prolonged exposure for N8, which lacks AChEi activity, might be clinically beneficial. The PK of primary metabolites mirrors that of buntanetap with fast plasma peak and clearance, while also exhibiting high brain absorption and a longer retention in human CSF [22].

To date, preclinical and clinical studies have utilized the anhydrous form of buntanetap. However, we have recently discovered a crystalline dihydrate form, which offers greater stability in the solid state without compromising its solubility. To distinguish between these forms, we will refer to the original anhydrous buntanetap as Form A and the new dihydrate crystalline form as Form B. The key difference between these forms lies in their hydration states. Form A is an anhydrate, meaning the lattice does not contain water. In contrast, Form B is a dihydrate with two moles of water in the crystal lattice. This fundamental difference makes the two forms distinct—both forms are stable and do not readily interconvert.

Previously, we have described the PK of Form A across several animal models and human studies, demonstrating achievement of fast plasma peak levels (within 2 h post-dose) and high brain tissue levels, followed by rapid clearance from plasma within 8 h and much longer clearance from CSF and brain within 12 h [22]. In this manuscript, we present our latest findings from studies comparing the PK profiles of Form A and Form B in mice, dogs, and humans.

## 2. Materials and Methods

### 2.1. Buntanetap Form A and Form B

#### 2.1.1. Synthesis of Form A (Anhydrate)

Buntanetap base was dissolved in ethanol (3.5 mL/g), and the solution was heated to 35–50 °C. A solution of D-tartaric acid (1.0 eq.) in ethanol (3.5 mL/g) and purified water (0.3 mL/g) was then added into the solution of buntanetap base at 35–50 °C. After stirring for 30 min, methyl t-butyl ether (MTBE, 12 mL/g) was then added at 35–50 °C and stirred for another 30 min. The batch was cooled to 5–20 °C and stirred for at least 30 min. The product was isolated by filtration, washed with 1.5:1 MTBE–ethanol mixture (4.5 mL/g), and dried at 60 °C under vacuum.

#### 2.1.2. Synthesis of Form B (Dihydrate)

Buntanetap base was dissolved in ethanol (5.0 mL/g), and the solution was heated to 50 °C. A solution of D-tartaric acid (1.1 eq.) in purified water (1.6 mL/g) was then added and stirred until a solution was obtained. The solution was cooled to 28 °C and seeded with Form B (0.5% *w*/*w*). After stirring for at least 1 h at 28 °C, MTBE (20 mL/g) was then added and stirred for at least another 1 h at 28 °C. The slurry was cooled to 0 °C and stirred for several hours at that temperature. The product was isolated by filtration and washed with 4 mL/g of a mixture of ethanol/water/MTBE: 0.75/0.25/3. The product was then dried at 40 °C under vacuum.

#### 2.1.3. Buntanetap Characterization

The polymorph screen was performed by Triclinic Labs (Lafayette, IN, USA), which identified and characterized Form B using x-ray powder diffraction (XRPD), differential scanning calorimetry (DSC), and thermogravimetric analysis (TGA). The crystal growth for XRPD analysis and interpretation was conducted by Pharmaron (Ningbo, China).

The XRPD was used to determine and characterize the crystal structure of Form A and Form B [23,24,25]. A Rigaku Smart Lab (Wilmington, MA, USA) X-ray diffractometer was configured in Bragg–Brentano reflection geometry equipped with a beam stop and a knife edge to reduce incident beam and air scatter. Data collection parameters are included in Appendix A.

The DSC analysis was carried out on a TA Instruments Q2000 Discovery Series (TA Instruments, New Castle, DE, USA) instrument [26,27]. The instrument temperature calibrations were performed using indium. The DSC cell was kept under a nitrogen purge of ~50 mL/min during the analysis. The sample was placed in a standard, crimped aluminum pan and heated from approximately 25 °C to 300 °C at a rate of 10 °C per min.

TGA analyses were carried out using a TA Instruments Q5500 Discovery Series instrument (TA Instruments, New Castle, DE, USA) [28,29]. The instrument balance was calibrated with class M weights, and the temperature calibration was performed with alumel. The nitrogen purge was ~10 mL/min at the balance and ~25 mL/min at the furnace. The sample was placed into a pre-tared platinum pan and heated from approximately 25 °C to 300 °C at a rate of 10 °C/min.

The dynamic vapor sorption (DVS) analysis was carried out using a TA Instruments Q5000 DVS analyzer (TA Instruments, New Castle, DE, USA). A sample was loaded into a metal-coated quartz pan and analyzed at 25 °C after being equilibrated to 5% relative humidity (RH) in 10% RH steps from 5 to 95% RH (adsorption cycle) and from 95 to 5% RH (desorption cycle). The movement from one step to the next occurred either after satisfying the equilibrium criterion of 0.01% weight change in 60 min or, if the equilibrium criterion was not met, after 90 min. The % weight change values were calculated in Microsoft Excel.

Buntanetap crystals were visualized using polarized light microscope (PLM, ECLIPSE LV100POL, Nikon, Tokyo, Japan). Form A only formed micro-crystals, which had to be visualized by micro-crystal electron diffraction (MicroED) and was further used for structure determination [30,31]. The instrument parameters are described in Appendix A. A small amount of sample was spread over a holey carbon electron microscope grid, placed on cryo-transfer holder, and inserted into Thermo Fisher Scientific Talos F200C electron microscope (Waltham, MA, USA). Crystals were illuminated with a parallel electron beam, and sequential diffraction frames per crystal were collected. The total rotation angle of micro-crystal was 80–110° with an exposure time of 0.5 s for single diffraction pattern. An initial structure model was obtained using intrinsic phasing algorithm in SHELXT and refined with SHELXL program (Version 2019/3). The atom type for all non-hydrogen atoms was determined by considering the electrostatic potential at each atom position, while taking into account the structural geometry, including bond length and assumed 2D structure. The identification of bond types in the structure was facilitated through an analysis of structural geometry.

### 2.2. Animal Models

#### 2.2.1. Mouse

Male mice (*Mus musculus*, CD1-ICR, *n* = 30), 7–9 weeks old, were supplied by Beijing Weitong Lihua Experimental Animal Technology Co., Ltd. (Vital River Laboratories, Bejing, China). The animals were confirmed to be healthy by WuXi AppTec (Shanghai, China) veterinarians before being assigned to the study. Mice received a single dose of buntanetap (65 mg/kg), either Form A or Form B, administered by oral gavage formulated as a homogeneous suspension in corn oil (no washout period). The environment controls were set to maintain a temperature range of 20–26 °C, a relative humidity range of 40 to 70%, and a 12-h light/12-h dark cycle. The light/dark cycle was interrupted for study-related activities. The administration volume was adjusted based on each animal’s weight on the dosing day (mean weight = 28.42 g). Prior to dosing, the mice were fasted. All experimental procedures were conducted in accordance with the WuXi AppTec IACUC (Protocol #PK01-001-2021 v1.4.) guidelines that are in compliance with the Animal Welfare Act, the Guide for the Care and Use of Laboratory Animals.

Blood was collected at 0.25, 0.5, 0.75, 1, 1.25, 1.5, 1.75, 2, 3, 4, 5, 6, 8, 10, and 12 h after dosing. Sample volume was 0.025 mL for each time point, collected via the saphenous vein. Samples were placed into a microcentrifuge tube containing the anti-coagulant K2-EDTA and centrifuged (3200× *g* for 10 min at 4 °C) within an hour of collection. For each form of buntanetap, a total of 3 plasma samples were acquired at every time point (Appendix A). At 10 h and 12 h post-dosing, buntanetap was below quantifiable limit (BQL), and hence these time points were not included in the analysis.

CSF was collected at 0.5, 1, 2, 6, and 12 h after dosing. Sample volume was 3 µL per time point, collected at terminal points following CO_2_ euthanasia via cisterna magna puncture, with the puncture site being the foramen magnum (no catheterization). Samples were frozen on dry ice and stored at −60 °C for further analysis. For each form of buntanetap, a total of 3 CSF samples were acquired at every time point (Appendix A). All CSF samples were transparent.

The whole brain was collected at 0.5, 1, 2, 6, and 12 h after dosing. For each form of buntanetap, a total of 3 samples were collected at every time point (Appendix A), following CO_2_ euthanasia. Samples were then washed in cold saline, dried, and homogenized immediately. Homogenizing buffer MeOH/15 mM PBS (1:2, *v*:*v*) was used at a ratio of 1:9. The tissue homogenate was kept at −70 ± 10 °C until LC-MS/MS analysis.

For samples collected within the first hour of dosing, ±1 min was acceptable, while for all other time points, samples were taken within 5% of the scheduled time and were not considered as protocol deviations, according to the vendor’s SOP. The narrow timing deviation ensured accuracy in capturing the peak drug concentration and the early stages of drug absorption, while also maintaining consistency among samples.

#### 2.2.2. Dog

Beagle dogs (*Canis lupus familiaris*, *n* = 12: male *n* = 6, female *n* = 6) were supplied by Jiangsu Marshall Biotechnology Co., Ltd. (Suzhou, China). The animals were confirmed to be healthy by WuXi AppTec veterinarians before being assigned to the study. Each animal was given a unique identification number marked on the ear and written on the cage card. The rooms were controlled and monitored for relative humidity (targeted range 40–70%) and temperature (targeted range 18–26 °C) and were on a 12-h light/dark cycle except when interruptions were necessitated by study activities. Fresh drinking water was available to all animals, ad libitum. Animals were fed twice daily except for the fasted animals with approximately 220 g of certified animal diet (Appendix A) from a certified vendor (Cooperative Medical Biological Engineering Co., Ltd. (Yangzhou City, China)). The Form A and Form B without excipients were manually filled into capsule shells which were manufactured from porcine gelatin (Torpac Inc., Mumbai, India), certified by USDA and processed according to regulation No. 853/2004 and compliant with 1069/2009 as amended by 142/2011 (Appendix A). All experimental procedures were approved by IACUC (Protocol #SZ20230512-Dogs) on 18 May 2023.

Dogs received a single oral dose of buntanetap (20 mg/kg) Form A or Form B in a cross-over design on study Day 1 and on study Day 15 following a 14-day washout period—one Torpac size #12 capsule per dog. Animals were weighed (mean weight = 8.47 kg) prior to dose administration on the day of dosing and fasted overnight before dosing on Day 1 and Day 15. Pre-dose concentrations on Day 15 were BQL, confirming the absence of carry-over during the washout period. Approximately 0.5 mL blood was collected at each time point (0.25, 0.5, 0.75, 1, 2, 4, 6, 8, and 12 h) via peripheral vessel from each study animal. The acceptable deviations on sampling time were ±1 min for the time points pre-dose through 1 h post-dose, and ±5% of the nominal time for time points after 1 h post-dose. Cocktail blood stabilizer was added to commercial tubes containing K2-EDTA at a blood sample ratio of 1:20 (*v*:*v*). The samples were placed on wet ice and processed for plasma within 60 min of collection by centrifugation at approximately 4 °C, 3200× *g* for 10 min. The plasma samples were divided into approximate 0.1 mL × 2 aliquots (one for BA, and the other one for back up) and transferred into labeled polypropylene microcentrifuge tubes. Then the samples were quickly frozen over dry ice and kept at −60 °C or lower until LC-MS/MS analysis.

### 2.3. Human Volunteers

This study was conducted in accordance with the ethical principles defined in the Declaration of Helsinki and in accordance with Good Clinical Practice (GCP), as delineated by Title 21 CFR Parts 50, 56, and 312, and ICH guidelines and directives, and was approved by Advarra Institutional Review Board (IRB), registered with Office for Human Research Protections (OHRP) and Food and Drug Administration (FDA) under IRB#00000971.

Healthy subjects were used for this study to allow for assessments in the absence of confounding factors such as comorbid conditions and concomitant medications. A total of 35 healthy male and female volunteers (Form A, *n* = 17; Form B, *n* = 18), 18 to 55 years of age with body mass index (BMI) ≥18.0 to ≤32.0 kg/m^2^ and body weight ≥50 kg, were enrolled in the study. Screening assessments included vital signs (blood pressure, pulse rate, respiratory rate, oral temperature), clinical laboratory testing, resting 12-lead electrocardiograms (ECGs), and physical examination. Subjects were excluded from participation if they used over-the-counter drugs within 7 days or any prescription drugs within 14 days prior to the first dose of study drug or received an investigational drug or participated in a clinical trial within 3 months prior to the first dose of the study drug.

Following a screening period of up to 28 days, eligible subjects were admitted to the clinical research unit (CRU) on Day-1. After an overnight fast of at least 10 h, each subject received their assigned treatment of buntanetap as a 50 mg oral capsule with approximately 240 mL (8 oz) of water. The current dose was chosen to ensure robust quantification of buntanetap and its metabolites in plasma. In this randomized, cross-over design, doses of Form A and Form B were separated by at least a 5-day washout period. Serial blood samples for PK analysis were collected from pre-dose through 24 h post-dose during each period. Safety was assessed by monitoring adverse events and screening parameters described above at various time points during the study. The intensity of AEs was assessed using the National Cancer Institute Common Terminology Criteria for Adverse Events (CTCAE), Version 5. Adverse events were coded using the most current version of the Medical Dictionary for Regulatory Activities (MedDRA) dictionary (Version 25.0 or higher).

### 2.4. Data Analysis

In animals, the concentrations of buntanetap and its N1- and N8-norbuntanetap metabolites were determined by using a validated LC-MS/MS method and subjected to a non-compartmental analysis (NCA). The linear/log trapezoidal rule was applied in obtaining the PK parameters. Individual plasma concentration values, which were below the lower limit of quantitation (LLOQ) that appear before Tmax, were set to zero and those after Tmax were excluded from the parameter calculation. The following PK parameters were calculated: Cmax, Tmax, T1/2, AUC0-last, and AUC0-inf. Collected data were analyzed using repeated measures mixed model analysis of variance and presented as mean values of each time point group. Descriptive statistics—sample size (n), arithmetic mean, standard deviation (SD), and coefficient of variation expressed as a percent (%CV)—were generated.

In humans, the PK of buntanetap and its metabolites in plasma was determined by NCA using Phoenix WinNonlin software (Version 8.1 or higher, Certara USA, Inc. Radnor, PA, USA) based on actual plasma sample collection time points. The following PK parameters were calculated: Cmax, Tmax, T1/2, AUC0-last, and AUC0-inf. A sample size of 18 was used to provide approximately 85% statistical power, with a 10% discontinuation rate for the comparison of PK parameters AUC0-last and Cmax. For the power calculation, the mean and standard deviation of the AUC0-last 95.5 (36.1) and Cmax 42.5 (11.1) were based on an internal parallel study of 50 mg Form A. It is assumed that Form A and Form B study drug are equivalent, with a correlation (intrasubject variability) of 0.15 (based on FDA guidance for Phase 1 BA/BE), bioavailability ratio of 1.5, and alpha of 0.05. A mixed-effect model (on log-transformation) that includes treatment, period, and treatment sequence as fixed effect, and subject nested within treatment sequence as a random effect were used to estimate the least square means and intrasubject variance. The 90% confidence intervals (CIs) for the ratios of geometric mean of AUC0-last, AUC0-inf, and Cmax were constructed to demonstrate the absorption and systemic exposure of each test versus reference comparison. Data are presented as means with SD.

## 3. Results

### 3.1. Buntanetap Polymorphism

Buntanetap Form B was discovered during a polymorph screen [32,33] as a dihydrate of Form A with two moles of water in the crystal lattice (Figure 1). Both forms are reproducibly crystallized from mixtures of water, ethanol, and antisolvent (e.g., MTBE). The crystallization outcome depends on water activity (aw): Form A is obtained when aw is below ~0.45 and Form B when aw exceeds this threshold. Both polymorphs are stable, do not readily interconvert, and can be distinguished by XRPD (Appendix A).

The two forms exhibit distinct physical properties: Form A melts at ~150.1 °C (as determined by DSC) and loses <1% of its weight before reaching its melting point, as measured by TGA (Appendix A). In contrast, Form B melts at ~93.3 °C, with a weight loss of ~7% prior to and during melting, consistent with the theoretical 6.9% water content of the dihydrate. In terms of stability, neither form was changed upon DVS, while Form B was also confirmed stable when dried overnight at 50 °C in a vacuum oven, showing no form conversion or water loss. The solubility of Form B was 14 mg/mL in water, which is a fraction of Form A solubility (>100 mg/mL). This is not unexpected since hydrates are typically less soluble in water than the corresponding anhydrous forms. While the apparent kinetic solubilities were different, Form A converts to Form B under physiological conditions, having no impact on absorption.

### 3.2. Buntanetap Metabolism

When metabolized, buntanetap (Form A and Form B alike) is converted into N1-norbuntanetap and N8-norbuntanetap (Figure 2) [20]. Like the drug itself, N8-norbuntanetap lacks AChEi activity, while N1- norbuntanetap exhibits weak AChEi activity, which may contribute to dose-limiting effects at high doses. In previous studies, human volunteers reported nausea and vomiting at 160 mg QD, which is eight times higher than the effective dose of 20 mg [3]. Overall, buntanetap has demonstrated a strong safety profile in over 800 participants enrolled in eleven clinical studies involving both healthy volunteers and patients with AD and PD with no serious drug-related adverse events [1,2,3].

### 3.3. Buntanetap PK in Mice

We first examined the distribution of buntanetap Form A and Form B, as well as its metabolites, in mouse plasma. Form B reached a Cmax of 828 ± 192 ng/mL at Tmax 0.25 h and a systemic exposure (AUC0-last) of 1186 ng·h/mL (Table 1, Figure 3a). With no significant difference (*p* > 0.05), Form A reached a Cmax of 939 ± 445 ng/mL at Tmax 0.5 h and an AUC0-last of 1707 ng·h/mL. The half-life (T1/2) was 0.56 h for Form A and 0.97 h for Form B. Due to the composite sampling in the current study, no SD variations were available for parameters other than Cmax.

N1-norbuntanetap was the least abundant metabolite, reaching a Cmax of 554 ± 275 ng/mL at Tmax 0.72 h for Form A and 468 ± 255 ng/mL at Tmax 0.25 h for Form B (Table 1, Figure 3b). The AUC0-last was 1403 ng·h/mL for Form A and 1312 ng·h/mL for Form B, with corresponding T1/2 values of 1.25 h and 1.94 h, respectively.

In contrast, N8-norbuntanetap was the most abundant metabolite, reaching a Cmax of 1931 ± 909 ng/mL at Tmax 1.5 h for Form A and 1425 ± 309 ng/mL at Tmax 1.75 h for Form B (Table 1, Figure 3c). The AUC0-last was 5334 ng·h/mL for Form A and 5598 ng·h/mL for Form B, with T1/2 values of 0.86 h and 1.32 h, respectively.

The Cmax ratios of metabolites to buntanetap for Form A were 0.59 ± 0.62 (N1/buntanetap) and 2.05 ± 2.04 (N8/buntanetap), and 0.56 ± 1.32 (N1/buntanetap) and 1.72 ± 1.61 (N8/buntanetap) for Form B.

Next, we assessed the ability of both buntanetap forms to penetrate the CNS by measuring concentrations in CSF and brain tissue, and by calculating their respective ratios to plasma concentrations. High brain concentrations are indicative of efficient CNS penetration, enabling the drug to reach its targets and exert therapeutic effects on cognition and function.

As anticipated, both forms showed substantial brain accumulation (Form A: 4737 ± 1973 ng/g; Form B: 2583 ± 239 ng/g), with brain/plasma ratios of 5.19 ± 0.625 for Form A and 4.64 ± 0.112 for Form B (Table 2). In contrast, concentrations in CSF were lower than in brain and plasma (Form A: 163 ± 87.8 ng/mL; Form B: 95.7 ± 7.45 ng/mL), with CSF/plasma ratios not exceeding 0.2 (Form A: 0.175 ± 0.0409; Form B: 0.186 ± 0.0278). Similar distribution patterns were observed for buntanetap’s metabolites. Notably, N8-norbuntanetap consistently emerged as the predominant species in both brain tissue (Form A: 7293 ± 3068 ng/g; Form B: 5508 ± 1409 ng/g) and CSF (Form A: 309 ± 131 ng/mL; Form B: 228 ± 5.77 ng/mL), exceeding levels of buntanetap.

### 3.4. Buntanetap PK in Dogs

In this part of the study, both male and female dogs were used. Our earlier research indicated no sex-related differences in buntanetap PK [22], and to confirm this, we first assessed potential sex-dependent variations in the current experiments. Consistent with previous findings, PK parameters showed no significant differences between sexes (Appendix A). The *p*-values of systemic exposure of both forms of buntanetap in male and female dogs according to *t*-test were Cmax (*p* = 0.695), T1/2 (*p* = 0.703), Tmax (*p* = 0.526), and AUC0-last (*p* = 0.545). No significant difference was also observed for the primary metabolites: N1—Cmax (*p* = 0.778), T1/2 (*p* = 0.680), Tmax (*p* = 0.603), AUC0-last (*p* = 0.430); and N8—Cmax (*p* = 0.686), T1/2 (*p* = 0.959), Tmax (*p* = 0.567), AUC0-last (*p* = 0.552). Therefore, the results are presented as sex-aggregated data.

For buntanetap Form B, Cmax reached 962 ± 480 ng/mL and Tmax 0.98 ± 0.55 h while AUC0-last was 1390 ± 500 ng·h/mL (Figure 4). These values are not significantly different from Form A (*p* > 0.05), which reached Cmax of 1040 ± 539 ng/mL, Tmax 1.13 ± 0.56 h, and AUC0-last 1530 ± 610 ng·h/mL. The T1/2 was 2.44 ± 0.69 h for Form A and 2.34 ± 0.54 h for Form B (Table 3).

The PK of primary metabolites was also consistent between Form A and Form B. In particular, for N1-norbuntanetap Cmax was 648 ± 151 ng/mL and 630 ± 153 ng/mL, with Tmax at 2.08 ± 0.67 h and 2.25 ± 0.87 h for Forms A and B, respectively. The AUC0-last was 2560 ± 1210 ng·h/mL for Form A and 2240 ± 636 ng·h/mL for Form B. The T1/2 was 1.48 ± 0.39 h for Form A and 1.54 ± 0.23 h for Form B.

For the N8-norbuntanetap, Form A and B had Cmax values of 99.5 ± 53.6 ng/mL and 90.3 ± 56.1 ng/mL, with Tmax at 1.15 ± 0.55 h and 1.02 ± 0.5 h, respectively. The AUC0-last was 235 ± 86.5 ng·h/mL for Form A and 215 ± 80.6 ng·h/mL for Form B. The T1/2 was 2.22 ± 0.75 h for Form A and 2.21 ± 0.58 h for Form B.

The Cmax ratios of metabolites to buntanetap for Form A were 0.62 ± 0.28 (N1/buntanetap) and 0.10 ± 0.09 (N8/buntanetap), and 0.65 ± 0.31 (N1/buntanetap) and 0.09 ± 0.12 (N8/buntanetap) for Form B.

Distribution of plasma concentration over a 12-h window showed similar profiles between Forms A and B and their metabolites, with both being fully cleared after 12 h post-dosing (Figure 4). Overall, none of the corresponding parameters showed significant differences between Forms A and B (*p* > 0.05).

### 3.5. Buntanetap PK in Humans

In this part of the study, human volunteers, both male and female, received 50 mg of buntanetap followed by the plasma PK evaluation over 24 h. Overall, mean (SD) age was 39.9 (8.43) years, the range of age was 24 to 51 years, and mean (SD) BMI was 26.90 (3.040) kg/m^2^. Racial composition included 10 (55.6) White and 8 (44.4) Black or African American subjects. A total of 11 (61.1%) subjects were Hispanic or Latino and 7 (38.9%) subjects were not Hispanic or Latino. One subject experienced a Grade 1 treatment emergent AE (TEAE) of back pain, that was considered unrelated to the study drug. There were no Grade 2 or higher TEAEs, no serious AE (SAEs), no deaths, and no treatment-related TEAEs. No subjects discontinued due to a TEAE.

For Form B, Cmax reached 71.1 ± 42.2 ng/mL and Tmax 1.5 h while AUC0-last was 153 ± 64.3 ng·h/mL (Figure 5). These values corresponded to those seen for Form A, which showed Cmax of 67.8 ± 33.5 ng/mL, Tmax 2.0 h, and AUC0-last 149 ± 47.9 ng·h/mL. The T1/2 was 3.28 ± 1.23 h for Form A and 3.54 ± 0.99 h for Form B (Table 4).

The PK of primary metabolites in humans was consistent between the two forms. N1-norbuntanetap reached Cmax of 7.12 ± 1.84 ng/mL and 6.91 ± 2.27 ng/mL, with Tmax at 2.48 h and 2.69 h for Forms A and B, respectively. For the N8- norbuntanetap, Form A and Form B had Cmax values of 7.87 ± 2.25 ng/mL and 7.25 ± 2.53 ng/mL, with Tmax at 2.58 h and 2.89 h, respectively. Similarly, the other parameters, such as AUC0-last, AUC0-inf, and T1/2 were almost equivalent between the polymorphs (Table 4).

The Cmax ratios of metabolites to buntanetap for Form A were 0.10 ± 0.05 (N1/buntanetap) and 0.12 ± 0.07 (N8/buntanetap) and 0.10 ± 0.05 (N1/buntanetap) and 0.10 ± 0.06 (N8/buntanetap) for Form B.

Bioequivalence of two forms in humans was assessed over a 24-h period, revealing identical profiles. Buntanetap, which reached higher concentrations than its metabolites, was fully cleared within 8 h, while N1 and N8 were eliminated by 24 h.

## 4. Discussion

The cross-study data presented in this manuscript provides an additional layer of understanding of structural, metabolic, and PK characteristics of buntanetap and introduces a novel crystal polymorph. Our discovery of Form B, a dihydrate of Form A, and a thorough comparative analysis of the two forms conducted in animals and humans represent a key step in adopting a crystal buntanetap in clinical studies for AD, PD, and other neurodegenerative conditions.

The two forms differ in the amount of water present during the synthesis process, with Form B having two moles of water in its lattice. We have characterized the physical properties of both forms and determined—based on differences in melting points and weight loss profiles as well as crystal structure determination—that Form B is distinct from Form A, and the two do not easily interconvert. In general, crystalline compounds offer a significant advantage in pharmaceutical development, as they tend to possess higher purity and greater resistance to physical instabilities [34]. Notably, Form B remains stable in its solid state even after drying, highlighting its suitability for use in various formulations without risk of conversion or degradation.

Bioequivalence studies demonstrated that Forms A and B exhibit similar PK properties and show efficient brain penetration. Both forms rapidly reach peak plasma concentration and are completely cleared within 12 h. Moreover, measurement of brain/plasma ratios in mice demonstrated that both forms achieve ~5-fold higher concentrations in the brain than in plasma, which is explained by the drug’s affinity for lipid tissue. Buntanetap is a lipophilic molecule with a partition coefficient (LogD) 2.2, which falls in the optimal range for BBB permeability, resulting in favorable CNS distribution with higher drug concentration in the brain. These findings align with previous PK data for Form A in several animal studies (mice, rats, dogs) and humans [22], where we observed extended half-life in CSF and brain, which is likely to contribute to improvements in cognitive function. In fact, our Phase 2/3 clinical trial in early AD patients (MMSE 21-24) revealed a statistically significant improvement in cognition (measured by ADAS-Cog11) by buntanetap in a dose-dependent manner (7.5, 15, and 30 mg) over 12 weeks (NCT05686044). Similarly, in the Phase 3 PD clinical study, buntanetap (10 and 20 mg) stopped cognitive decline in the ITT (intent-to-treat) population of early PD after 24 weeks of treatment and improved cognition in patients with mild cognitive deficits (MMSE 20-26) (NCT05357989).

Buntanetap is metabolized into two primary metabolites: N1- and N8-norbuntanetap. Our data confirms this metabolic pathway with no notable differences in metabolite profiles between the two polymorphs. In humans, plasma concentrations of both N1 and N8 are significantly lower than those of buntanetap, and N8 shows a longer half-life than N1. Since N8 lacks the activity associated with AChE inhibition but retains the ability to inhibit neurotoxic proteins [5,18,20], its extended half-life may account for durable therapeutic effect without contributing to potential toxicity. Interestingly, in dogs the N1 metabolite is more prominent while in mice it is N8, for both forms. This difference is likely due to species-specific variations in CYP-mediated metabolism, as we discussed in our previous publication [22]. The CYP3A enzyme subfamily, which primarily drives buntanetap metabolism, shows notable interspecies differences in sequence homology and catalytic function. In humans, CYP3A4 is the main enzyme responsible for buntanetap metabolism. In mice, the closest analog is CYP3A11, sharing 70–75% sequence identity, and in dogs it is CYP3A12 with ~79% homology [35,36]. Despite these similarities, differences in substrate specificity and catalytic activity between species can lead to variations in the metabolic rate and profile of buntanetap. It is also important to highlight that in mice, N8 metabolite was observed at higher quantities than buntanetap, which might be attributed to species-specific differences in CYP-mediated metabolism, and specifically that mice generally exhibit a much faster first-pass metabolism by liver enzymes. As a result, the original drug is depleted quicker, and the preferred metabolite becomes dominant instead. Further studies are needed to better understand the interspecies differences in buntanetap’s metabolism. Despite these slight variations between mice, dogs, and humans, the metabolic profiles of Form A and Form B remained consistent within each species, which was the focus of the current study.

Data presented in this manuscript is important for supporting the use of Form B for clinical application; however, there are several limitations worth acknowledging. First, the study had a relatively small sample size and was limited to oral administration. Increasing the number of subjects and conducting the analysis of buntanetap via intravenous (IV) administration could provide more robust data on absolute bioavailability. However, the focus was to assess and compare the PK of two forms via a clinically relevant route of administration, consistent with the drug’s use as an oral dosage form in clinical trials. Second, while the efficacy data is currently available only for Form A, which is in line with observed BBB penetration and PK data, we are still in the process of gaining such data for Form B. Hence, our extrapolation of the possible link between the observed bioequivalence of two forms and the efficacy of Form B is speculative, and more data needs to be obtained to draw a confirmative conclusion.

Taken together, our findings support the selection of Form B for human clinical trials. Currently, a pivotal Phase 3 study in early AD (MMSE 21-28) is underway, in which patients receive 30 mg of buntanetap Form B, being conducted under an investigational new drug application approved by the FDA (NCT06709014). This Phase 3 study will include a 6-month symptomatic read-out and an 18-month disease-modifying read-out. Moreover, the patent claims for the composition of matter, manufacturing, mechanism of action, and the use of Form B for acute and chronic indications as well as in combination therapies have been filed for Form B, covering intellectual property protection till late 2040s.

## 5. Conclusions

The purpose of this study was to characterize the PK of a newly discovered crystalline polymorph of buntanetap, Form B, and to demonstrate that it exhibits comparable bioequivalence to Form A. Within each species, the key PK parameters remained remarkably consistent between the two forms, as well as between the primary metabolites. These findings support Form B as a candidate for progression in future clinical trials targeting AD, PD, and other neurodegenerative conditions.

## 6. Patents

The work reported in the manuscript resulted in the filing of the following patents: Composition of matter for Form B (US Provisional Application No. 63/509,356, filed 21 June 2023 (original filing); US Provisional Application No. 63/580,011, filed 1 September 2023 (added data re: substantially pure Form B); PCT Application No. PCT/US24/34966, filed 21 June 2024) and Methods of Manufacturing for Form B (US Provisional Application No. 63/656,876, filed 6 June 2024; PCT Application No. PCT/US25/32470, filed 5 June 2025).

## Figures and Tables

**Figure 1 biomolecules-15-01299-f001:**
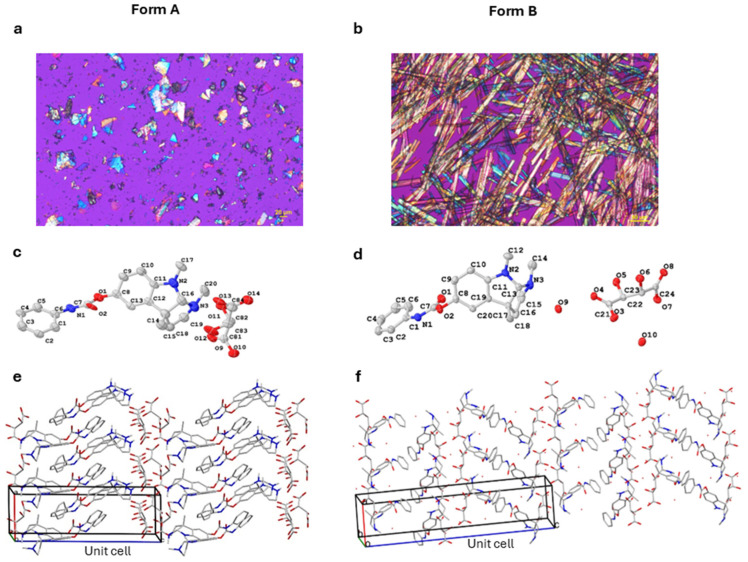
Photomicrograph of crystal structures of Form A (**a**) and Form B (**b**) using PLM. Stereochemistry of a single molecule obtained by MicroED is shown for Form A (**c**) and Form B (**d**). The atoms color coding as follows: C (gray), N (blue), O (red). Packing diagram of crystal structures for Form A (**e**) and Form B (**f**). Each unit cell contains four asymmetric crystal molecules. For images (**c**–**f**), hydrogen atoms were omitted for clarity.

**Figure 2 biomolecules-15-01299-f002:**
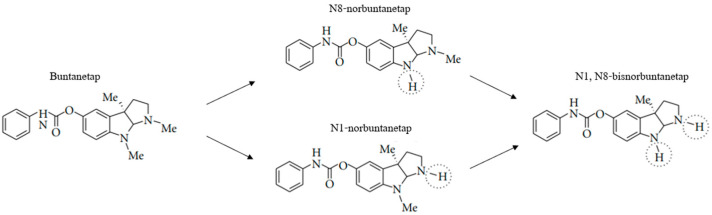
Schematic illustration of buntanetap metabolism into its primary metabolites.

**Figure 3 biomolecules-15-01299-f003:**
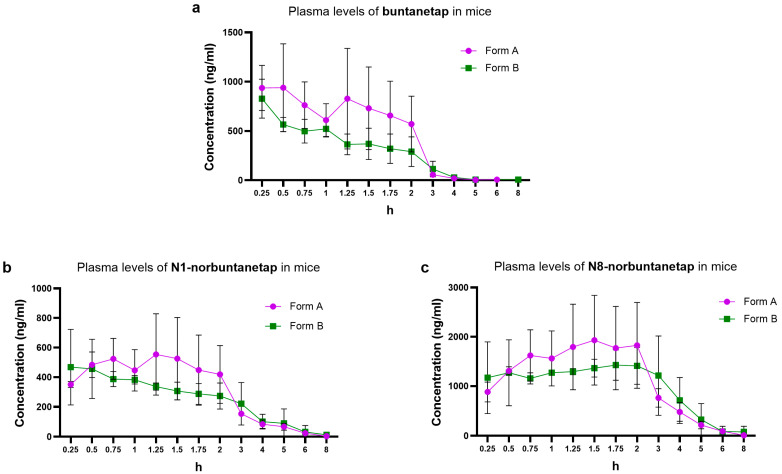
Plasma PK of buntanetap (**a**), N1-norbuntanetap (**b**), and N8- norbuntanetap (**c**) in CD-1 mice (65 mg/kg) over 8 h for Form A (*n* = 15) and Form B (*n* = 15).

**Figure 4 biomolecules-15-01299-f004:**
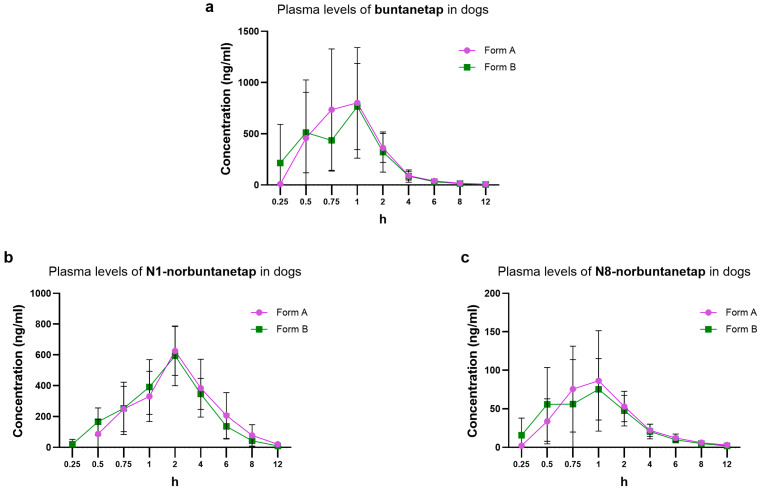
Plasma PK of buntanetap (**a**), N1-norbuntanetap (**b**), and N8-norbuntanetap (**c**) in Beagle dogs (20 mg/kg) for Form A (*n* = 12) and Form B (*n* = 12).

**Figure 5 biomolecules-15-01299-f005:**
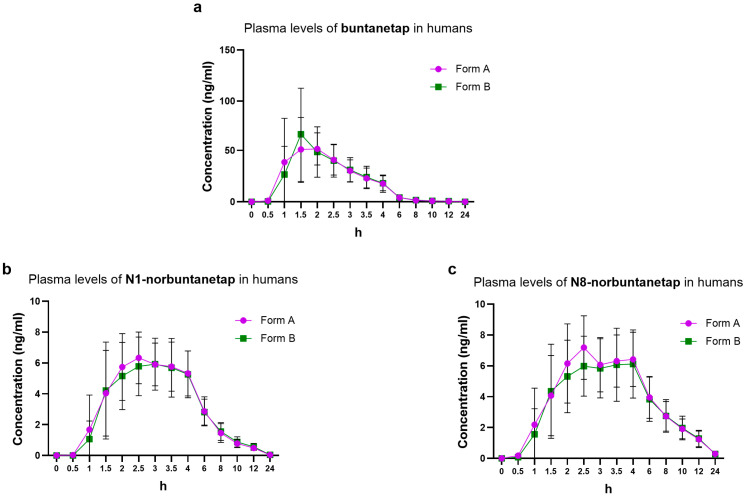
Plasma PK of buntanetap (**a**), N1-norbuntanetap (**b**), and N8- norbuntanetap (**c**) in humans (50 mg) for Form A (*n* = 17) and Form B (*n* = 18).

**Table 1 biomolecules-15-01299-t001:** Summary of PK parameters of plasma buntanetap Form A (*n* = 15) and Form B (*n* = 15) and its metabolites in CD-1 mice (65 mg/kg).

PK Parameters	Form A, *n* = 15	Form B, *n* = 15
	Buntanetap (65 mg/kg, Oral Gavage)	
Cmax	939 ± 445 ng/mL	828 ± 192 ng/mL
Tmax	0.5 h	0.25 h
T1/2	0.56 h	0.97 h
AUC0-last	1707 ng·h/mL	1186 ng·h/mL
AUC0-inf	1712 ng·h/mL	1195 ng·h/mL
	**N1-norbuntanetap**	
Cmax	554 ± 275 ng/mL	468 ± 255 ng/mL
Tmax	0.72 h	0.25 h
T1/2	1.25 h	1.94 h
AUC0-last	1403 ng·h/mL	1312 ng·h/mL
AUC0-inf	1407 ng·h/mL	1323 ng·h/mL
	**N8-norbuntanetap**	
Cmax	1931 ± 909 ng/mL	1425 ± 309 ng/mL
Tmax	1.5 h	1.75 h
T1/2	0.86 h	1.32 h
AUC0-last	5334 ng·h/mL	5598 ng·h/mL
AUC0-inf	5338 ng·h/mL	5624 ng·h/mL

**Table 2 biomolecules-15-01299-t002:** Distribution of buntanetap (65 mg/kg) and its metabolites in brain (ng/g) and CSF (ng/mL), and respective concentration ratios to plasma levels (ng/mL) in CD-1 mice.

	Brain (Cmax)	Brain/Plasma	CSF (Cmax)	CSF/Plasma
**Form A**				
Buntanetap	4737 ± 1973 ng/g	5.19 ± 0.625	163 ± 87.8 ng/ml	0.175 ± 0.0409
N1	1854 ± 476 ng/g	4.22 ± 0.345	54.4 ± 15.6 ng/ml	0.113 ± 0.0239
N8	7293 ± 3068 ng/g	4.37 ± 0.296	309 ± 131 ng/ml	0.206 ± 0.0395
**Form B**				
Buntanetap	2583 ± 239 ng/g	4.64 ± 0.112	95.7 ± 7.45 ng/ml	0.186 ± 0.0278
N1	1609 ± 733 ng/g	3.91 ± 0.303	43.2 ± 18.8 ng/ml	0.112 ± 0.0113
N8	5508 ± 1409 ng/g	4.35 ± 0.906	228 ± 5.77 ng/ml	0.188 ± 0.0363

**Table 3 biomolecules-15-01299-t003:** Summary of PK parameters of plasma buntanetap Form A (*n* = 12) and Form B (*n* = 12) and its metabolites in Beagle dogs (20 mg/kg).

PK Parameters	Form A, *n* = 12	Form B, *n* = 12
	Buntanetap (20 mg/kg, PO)	
Cmax	1040 ± 539 ng/mL	962 ± 480 ng/mL
Tmax	1.13 ± 0.56 h	0.98 ± 0.55 h
T1/2	2.44 ± 0.69 h	2.34 ± 0.54 h
AUC0-last	1530 ± 610 ng·h/mL	1390 ± 500 ng·h/mL
AUC0-inf	1560 ± 612 ng·h/mL	1410 ± 506 ng·h/mL
	**N1-norbuntanetap**	
Cmax	648 ± 151 ng/mL	630 ± 153 ng/mL
Tmax	2.08 ± 0.67 h	2.25 ± 0.87 h
T1/2	1.48 ± 0.39 h	1.54 ± 0.23 h
AUC0-last	2560 ± 1210 ng·h/mL	2240 ± 636 ng·h/mL
AUC0-inf	2610 ± 1290 ng·h/mL	2260 ± 645 ng·h/mL
	**N8-norbuntanetap**	
Cmax	99.5 ± 53.6 ng/mL	90.3 ± 56.1 ng/mL
Tmax	1.15 ± 0.55 h	1.02 ± 0.5 h
T1/2	2.22 ± 0.75 h	2.21 ± 0.58 h
AUC0-last	235 ± 86.5 ng·h/mL	215 ± 80.6 ng·h/mL
AUC0-inf	246 ± 91.4 ng·h/mL	221 ± 82.6 ng·h/mL

**Table 4 biomolecules-15-01299-t004:** Summary of PK parameters of plasma buntanetap Form A (*n* = 17) and Form B (*n* = 18) and its metabolites in humans (50 mg).

PK Parameters	Form A, *n* = 17	Form B, *n* = 18
	Buntanetap (50 mg, PO)	
Cmax	67.8 ± 33.5 ng/mL	71.1 ± 42.2 ng/mL
Tmax	2.0 h	1.5 h
T1/2	3.28 *±* 1.23 h	3.54 ± 0.99 h
AUC0-last	149 ± 47.9 ng·h/mL	153 ± 64.3 ng·h/mL
AUC0-inf	156 ± 43.3 ng·h/mL	163 ± 61.3 ng·h/mL
	**N1-norbuntanetap**	
Cmax	7.12 ± 1.84 ng/mL	6.91 ± 2.27 ng/mL
Tmax	2.48 h	2.69 h
T1/2	3.26 ± 0.36 h	3.23 ± 0.38 h
AUC0-last	33.6 ± 8.58 ng·h/mL	33.4 ± 9.68 ng·h/mL
AUC0-inf	33.8 ± 8.66 ng·h/mL	33.6 ± 9.74 ng·h/mL
	**N8-norbuntanetap**	
Cmax	7.87 ± 2.25 ng/mL	7.25 ± 2.53 ng/mL
Tmax	2.58 h	2.89 h
T1/2	5.48 ± 0.71 h	5.23 ± 0.72 h
AUC0-last	50.0 ± 14.5 ng·h/mL	48.2 ± 16.6 ng·h/mL
AUC0-inf	52.3 ± 15.1 ng·h/mL	51.4 ± 17.1 ng·h/mL

## Data Availability

All reports containing data are available in the reference section of the paper or upon request from Annovis Bio.

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
