# Peer review of "Pharmacokinetics of Novel Crystalline Buntanetap in Mice, Dogs, and Humans"

_biomolecules, 2025, doi:10.3390/biom15091299_

Round 1
Reviewer 1 Report
Comments and Suggestions for Authors
In this article a new form of buntanetap is characterized. The information included in the article is relevant and I only have a few comments about it.
I believe that the methodology for the experiments carried out in mice need to be further explained:
- Which form of buntanetap was received by each mouse?
- Was there a washout period as in the other models?
- How many replicates were used for the calculation of the Brain/Plasma ratio? If the whole brain was collected at 0.5, 1, 2, 6 and 12 h after dosing, 5 brain samples were taken and, as there were only 7 mice in the study, is that meaning that only one replicate was used in each time?
Author Response
Dear Reviewer,
We appreciate your comments regarding the relevance of our information and your valuable feedback. Please find attached our point-by-point responses to each of your remarks.
Best

Reviewer 2 Report
Comments and Suggestions for Authors
Major Comments
1.- Ethical aspects in humans and animals
Ethical approval is briefly mentioned at the end of the manuscript but not detailed in Materials and Methods.
-
Animals: Please include approval number, competent authority, and guidelines followed (e.g., Guide for the Care and Use of Laboratory Animals, EU Directive 2010/63/EU).
-
Humans: Please provide Institutional Review Board approval number, regulatory authority (FDA, AEMPS), and EudraCT code (or equivalent).
It is recommended to move this information into the Methods section to comply with journal and regulatory standards.
2.- Characterization of animals and humans
The description of animals and human participants lacks essential details:
-
Animals: Indicate taxonomic nomenclature and strain (e.g., Mus musculus Crl:CD1(ICR); Canis lupus familiarisBeagle), age, mean weight, housing conditions (temperature, humidity, light/dark cycle, diet, enrichment) and vendor.
-
Humans: Specify race/ethnicity, sex distribution, weigh mean, inclusion/exclusion criteria, and screening details (physical exam, lab tests, ECG, lifestyle factors).
3.- Mouse sampling design
The study uses n=7 male mice with multiple blood draws (13 time points) and CSF sampling (5 time points). This raises methodological concerns:
-
Total blood volume extracted vs ethical limits.
-
CSF sampling technique (cisterna magna, repeated puncture, serial sacrifice).
Please clarify whether different animals were used at each time point, total blood volumes, and method for CSF collection.
4.- Dog study: sex analysis and washout period
The manuscript states there were no sex-related differences, but no statistical results are provided. A formal analysis (ANOVA with sex as factor or stratified analysis) is needed.
The 14-day washout period should be justified with pre-dose concentration data confirming absence of carry-over.
5.- Animal diet
Lines 147–149 indicate “...220 grams of certified animal diet from a certified vendor.” but do not specify product name, manufacturer, or composition.
Diet composition (fat, protein, carbohydrate) can impact absorption of lipophilic drugs such as buntanetap. Please provide these details in line with ARRIVE guidelines.
6.- Capsule formulation in dogs
Line 152 indicates “Torpac size #12 capsule” but the internal formulation is not described. Please provide:
-
Excipients and proportions.
-
Filling process and capsule material.
-
Quality control criteria (uniformity of mass/content, pharmacopeia compliance).
7.- Human study: screening and regulatory authorizations
The phase I human study is briefly described. Please provide:
-
Screening procedures (medical history, physical examination, laboratory analysis, ECG, exclusion criteria).
-
Regulatory authorizations (IRB approval number, regulatory authority, EudraCT code).
This information is mandatory for phase I clinical trials.
8.- Confusion between bioavailability and bioequivalence (Key Comment)
The manuscript uses “bioavailability” to describe comparison of Form A vs Form B, which is conceptually incorrect:
-
Bioavailability requires IV comparison.
-
Bioequivalence compares PK between oral solid forms.
No formal bioequivalence analysis is presented (90% CI within 80–125%, ANOVA log-transformed data). This conceptual error undermines the main conclusion.
9.- Interpretation of plasma vs CSF clearance (Key Comment)
The finding that plasma clearance > CSF clearance is expected for lipophilic CNS-penetrant molecules and is not a novel observation. Without IV administration, Vd cannot be calculated and compartmental distribution cannot be modeled. This statement should be reframed to avoid presenting a basic PK phenomenon as a novel result.
10.- Discussion: extrapolation to clinical efficacy
The discussion extrapolates PK results of Form B to therapeutic efficacy based on clinical trial data from Form A. There are no efficacy data for Form B, making this extrapolation premature.
11.- Discussion: omission of limitations
Key study limitations are not acknowledged: small sample size, absence of IV, single-dose design in humans, and direct extrapolation to clinical use.
12.- Overstated conclusion
The conclusion states that Form B “retains therapeutic efficacy.” Without direct clinical data, this should be reformulated as: “Form B shows comparable PK to Form A and is a candidate for progression to clinical trials.”
13.- Absence of IV in humans (Key Comment)
The lack of IV administration prevents calculation of absolute bioavailability and limits interpretation of Cmax and AUC. Observed concentrations may reflect assay sensitivity without reference to concentrations required for therapeutic effect.
14.- Human dose and physicochemical characterization
Phase 2/3 AD trials use 10–20 mg; this phase I PK study uses 50 mg without justification. Please explain the rationale.
Lines 244–352 mention physicochemical differences between Forms A and B but provide no numerical data (solubility, melting point, stability). These are essential for interpreting PK and bioequivalence.
15.- Confusing interpretation of N1 and N8
Lines 373–376 attribute prolonged effect to N8 (inactive) due to its long half-life. This is incorrect: N8 does not contribute to efficacy. Short half-life of N1 (active) may reduce toxicity but does not relate to N8. Reformulation is recommended.
16.- Regulatory implications of active metabolite N1
Presence of active metabolite N1 requires additional PK, toxicity, and efficacy studies per regulatory guidance. This is not discussed, despite impacting phase III development complexity.
17.- CYP3A4 interactions and safety
Metabolism by CYP3A4 raises potential for drug–drug interactions and cardiac safety issues (QTc) in polytreated populations. While not the focus of this study, this should be acknowledged.
18.- PK tables
Specify whether Cmax, Tmax, AUC values are arithmetic/geometric means or medians. Include 90% CIs for Form A vs Form B. Ensure values are consistent between text and tables.
19.- N8 in mice and interspecies differences
In mice, N8 plasma concentrations exceed parent drug levels, and Cmax >1000 ng/mL vs ~50 ng/mL in humans. This should be discussed, with possible explanations (dose, metabolism, biodisposition).
20.- Marked concentration difference mouse vs human
The marked interspecies difference in plasma concentrations warrants discussion of PK/PD implications for dose extrapolation to humans.
21.- Brain/plasma and CSF/plasma comparison (Key Comment)
Brain concentrations are from total homogenate (free + bound), plasma is free concentration. This methodological difference invalidates direct comparison. Lipophilicity may explain apparent high brain concentration without indicating greater active fraction.
22.- Conflict of interest and ethical transparency
The manuscript presents indications of potential industry involvement, as buntanetap is developed by Annovis Bio. Authors in related publications are affiliated with Annovis Bio or contracted CROs. This should be explicitly declared in the Conflicts of Interest and Funding sections, specifying:
-
Author affiliations (academic, industrial, or CRO).
-
Role of Annovis Bio in study design, data collection, analysis, and manuscript preparation.
Additionally, ethical approvals require greater transparency:
-
Animals: Approval number, ethics committee, and guidelines followed (ARRIVE, EU Directive 2010/63/EU).
-
Humans: IRB/CEIm approval number, regulatory authority authorization (FDA/AEMPS), and EudraCT or IND code.
Without these details, the manuscript does not meet Biomolecules’ ethical and transparency standards.
Minor Comments
-
Scientific terminology: Revise language to improve conceptual precision (bioavailability vs bioequivalence, clearance interpretation, metabolite discussion).
-
Figures: Replace Y-axis labels “ng/mL” with “Concentration (ng/mL)”; expand legends to include species, dose, route, formulation, n.
-
Tables: Harmonize units; check consistency of values between text and tables.
-
Supplementary material: Include physicochemical properties of Form A/B (solubility, stability, melting point).
Author Response

(The authors gave the same response as above.)

Reviewer 3 Report
Comments and Suggestions for Authors
The article contains a lot of interesting information, but the authors should add a few details.
- In the introduction, the authors should provide more information on the PK of the drug and its metabolites, since the article covers the PK of the drug. Information on the pharmacological activity of the metabolites is missing.
- "administered by oral gavage (PO)" - this is not a valid abbreviation
- There is no information on the number of blood samples taken from mice. The authors wrote that they took 3 µl of CSF, but did not specify how the CSF was obtained, what anaesthesia was used, what type of cannula was used, where the puncture site was located, or whether all samples were transparent.
- Why was there a gender division in dogs but not in mice? Is it worth adding information about the influence of gender on PK?
- When examining PK in healthy volunteers, how were adverse reactions assessed, according to what criteria and scale? The severity of adverse reactions was not specified.
- When presenting concentrations in brain homogenates, information on residual brain blood is missing. (Fridén M, Ljungqvist H, Middleton B, Bredberg U, Hammarlund-
Udenaes M. Improved measurement of drug exposure in the brain
using drug-specific correction for residual blood. J Cereb Blood
Flow Metab. 2010;30:150–61. https ://doi.org/10.1038/jcbfm
.2009.200.) - The following must also be calculated: drug targeting index (DTI).
- The ratio of metabolites to the parent drug must be calculated (e.g. Pharmacokinetic Drug Interaction Study of Sorafenib and Morphine in Rats). If the metabolites exhibit pharmacological activity, the sum of the drug and metabolites must also be compared.
Author Response

(The authors gave the same response as above.)

Reviewer 4 Report
Comments and Suggestions for Authors
Overall Summary
This study demonstrates through rigorous cross-species bridging trials that Buntanetap’s new polymorph (Form B) maintains the pharmacological efficacy of Form A while offering enhanced stability, highlighting its clear translational value. Revision suggestions focus on enhancing data rigor (statistical methods/unit standardization), deepening discussion (metabolite functions/species variability), and ensuring precision (figures/patents) to strengthen academic robustness and industrial applicability.
Revision Suggestions
- Results and Discussion Section
- Clarify Statistical Differences:
Some PK parameters (e.g., mouse Form A `Cₘₐₓ = 939 ± 445 ng/mL` vs. Form B `828 ± 192 ng/mL`) are described as having "no significant differences" but lack p-values or statistical methods (e.g., ANOVA). Recommend adding statistical analysis details.
The prototype drug parameters of Form A and Form B showed no significant differences, but the kinetics and brain tissue distribution of their metabolites exhibited more pronounced variations. The p-values need to be supplemented, and the underlying reasons for these differences should be discussed.
- Metabolite Functional Annotation:
The N1-metabolite exhibits weak acetylcholinesterase inhibitory (AChEi) activity (potentially causing high-dose side effects), but the therapeutic implications of the N8-metabolite lacking this activity are unexplained. Recommend elaborating on their differential impact on efficacy/safety.
- Species Variability Analysis:
Dogs show higher N1-metabolite levels, while mice exhibit predominance of the N8-metabolite (possibly due to CYP enzyme species differences). Recommend discussing implications for cross-species extrapolation.
- Methodological Details
- Polymorph Synthesis Steps:
The solvent volume unit "vol." is undefined (e.g., does it mean mL/g?). Recommend standardizing to explicit units (e.g., "ethanol (5.0 mL/g)").
- Animal Study Sampling Time Deviation:
A ±1-minute deviation for sampling within 1 hour is mentioned without justification. Recommend citing relevant PK guidelines (e.g., FDA Bioanalytical Method Validation).
- Figures and Data
- Supplementary Figure Nomenclature:
The main text references "Supplementary Figure 3/4/5," while supplementary materials use "Figure S3/S4/S5." Recommend unifying to "Figure S1/S2/S3" format throughout.
- Figure 3–5 Labeling:
"Form B" is mislabeled as "Pom B" in Figure 3–5 (Page 10). Requires correction.
- Human PK Data Table:
The unit "ng-h/ml" in Table 4 should be standardized to "ng·h/mL" (consistent with Tables 1/3).
- Discussion and Conclusion
- Patent Information Timeliness:
A PCT application (PCT/US24/34966) dated June 2024 is noted, but the text states data supports "Form B for future trials." Recommend clarifying patent status (e.g., grant number).
- Water Activity (a_w) Threshold:
Formation of Form B at a_w >0.45 is stated without specifying measurement conditions (temperature/solvent). Recommend adding methodological details or references.
- Formatting and Language
- Abbreviation Definition:
Some abbreviations (e.g., LogD, a_w) lack definitions at first use (LogD appears abruptly in Discussion). Recommend defining all abbreviations in the Abstract/Introduction.
- Ethics Review:
Critical details are missing in the descriptions of animal ethics and patient informed consent (e.g., committee name, approval number, animal welfare measures, privacy protection methods). This may compromise the transparency and traceability of the ethics review. Detailed information should be provided.
- Reference Consistency:
For references, please cite cutting-edge articles to increase the scientific and forward-looking nature of the articles.
Reference [3] cites "Maccecchini et al." in-text but lists "Maccecchini, M.L." (without "et al.") in the bibliography. Requires formatting unification.

Author Response

(The authors gave the same response as above.)

Round 2
Reviewer 2 Report
Comments and Suggestions for Authors
I would like to sincerely thank you for your comprehensive and thoughtful responses to my previous comments. I appreciate the effort you invested in addressing the ethical, methodological, and conceptual issues raised during the initial review.
Your clarifications and manuscript revisions have significantly improved the quality and transparency of the work. I believe this study provides relevant pharmacokinetic data that will be of interest to researchers in the field.
Congratulations on the progress made, and I wish you all the best with the publication and future research endeavors.
Kind regards,
Reviewer 4 Report
Comments and Suggestions for Authors
The authors have comprehensively addressed all the reviewers' comments, and the revisions are scientifically rigorous with meticulous attention to detail. The manuscript demonstrates significant improvements in key areas such as pharmacokinetic mechanisms, species extrapolation, and metabolite activity. Formatting and ethical compliance issues have been thoroughly resolved, meeting the journal's publication standards.